# Transcriptome and Metabolome Dynamics Explain Aroma Differences between Green and Red Prickly Ash Fruit

**DOI:** 10.3390/foods10020391

**Published:** 2021-02-10

**Authors:** Xitong Fei, Yichen Qi, Yu Lei, Shujie Wang, Haichao Hu, Anzhi Wei

**Affiliations:** 1College of Forestry, Northwest Agriculture and Forestry University, Xianyang 712100, China; feixitong@nwafu.edu.cn (X.F.); qiyichen@nwafu.edu.cn (Y.Q.); leiyu1997@nwafu.edu.cn (Y.L.); wangshujie@nwafu.edu.cn (S.W.); huhaichao@nwafu.edu.cn (H.H.); 2Research Centre for Engineering and Technology of Zanthoxylum State Forestry Administration, Yangling, Xianyang 712100, China

**Keywords:** fruit aroma, terpenoid synthesis, *Zanthoxylum bungeanum*, MVA pathway, MEP pathway

## Abstract

Green prickly ash (*Zanthoxylum armatum*) and red prickly ash (*Zanthoxylum bungeanum*) fruit have unique flavor and aroma characteristics that affect consumers’ purchasing preferences. However, differences in aroma components and relevant biosynthesis genes have not been systematically investigated in green and red prickly ash. Here, through the analysis of differentially expressed genes (DEGs), differentially abundant metabolites, and terpenoid biosynthetic pathways, we characterize the different aroma components of green and red prickly ash fruits and identify key genes in the terpenoid biosynthetic pathway. Gas chromatography-mass spectrometry (GC-MS) was used to identify 41 terpenoids from green prickly ash and 61 terpenoids from red prickly ash. Piperitone was the most abundant terpenoid in green prickly ash fruit, whereas limonene was most abundant in red prickly ash. Intergroup correlation analysis and redundancy analysis showed that *HDS2*, *MVK2*, and *MVD* are key genes for terpenoid synthesis in green prickly ash, whereas *FDPS2* and *FDPS3* play an important role in the terpenoid synthesis of red prickly ash. In summary, differences in the composition and content of terpenoids are the main factors that cause differences in the aromas of green and red prickly ash, and these differences reflect contrasting expression patterns of terpenoid synthesis genes.

## 1. Introduction

Terpenoids are among the most structurally diverse and abundant plant secondary metabolites, and more than 50,000 terpenoids have been discovered [1]. They are usually produced in fruits and flowers but are also found in leaves, stems, and roots [2]. The general formula of terpenoids is (C_5_H_8_)*_n_*, where *n* is the number of isoprene units. Based on this number, terpenoids are classified as semiterpenes (C_5_), monoterpenes (C_10_), sesquiterpenes (C_15_), diterpenes (C_20_), disesquiterpenes (C_25_), triterpenes (C_30_), tetraterpenes (C_40_), and polyterpenes (C > 40) [3]. Although plant terpenoids vary greatly in structure, they all derive from the same C5 isoprene skeleton that is produced from isopentenyl pyrophosphate (IPP) and dimethylallyl pyrophosphate (DMAPP) and is used to synthesize the terpenoid backbone through rearrangement and cyclization reactions. Terpenoid synthesis involves two different biosynthetic pathways, the mevalonate (MVA) pathway, and the 2-C-methyl-D-erythritol-4-phosphate (MEP) pathway, also known as the 1-deoxy-D-xylosyl-5-phosphate (DXP) pathway [4,5]. Monoterpenes, diterpenes, tetraterpenes, zeatin, ubiquinone, and other terpenoid quinones are synthesized primarily through the MEP pathway, whereas sesquiterpenes and triterpenoids are synthesized primarily through the MVA pathway [6,7]. Based on the number of carbon rings in their molecular structures, terpenoids can be divided into chain terpenes, monocyclic terpenes, cyclic terpenes, bicyclic terpenes, tetracyclic terpenes, and so on [8]. In addition, many terpenoids are also oxygen-containing derivatives and can therefore be divided into alcohols, acids, ketones, carboxylic acids, esters, and glycosides [9,10].

Many terpenoids are highly specific to different plant species and play an important role in signal transduction, abiotic and biotic stress resistance, biological interactions, etc. [11,12]. For example, monoterpenoids such as nerol and sesquiterpenoids such as farnesene and caryophyllene are volatiles that can exert toxic effects on herbivorous insects and pathogenic microorganisms. As signal molecules, terpenoids can also attract natural enemies of insects and play an active part in the protection of plants from insects and pathogens [13,14,15,16]. In tea plants, terpenoids function as signals among plants during low temperature stress. Low temperature can induce tea plants to produce nerolidol, which other tea plants absorb from the air and convert into a glucoside, thereby enhancing their low temperature tolerance [17]. Terpenoids are widely used in medicine and exhibit antioxidant, antibacterial, anti-inflammatory, and anti-tumor activity [18,19,20,21]. In addition, terpenoids influence the flavor characteristics and quality of fruits and are the main components of fruit aroma. In sweet orange, ethyl butanoate and myrcene are the main aroma components of fruit, and limonene and myrcene are the main aroma components of peel [22,23]. Strawberry has abundant aroma components including esters, terpenes, alkanes, and alcohols; trans-nerolidol (a sesquiterpene) is released at particularly high levels [24]. Sixty-seven free volatiles and 79 bound volatiles were identified in kiwi fruit and wine. Alcohol was the most abundant free volatile, and terpenoids were characteristic bound volatiles [25]. Headspace solid-phase microextraction (SPME) and GC combined with mass spectrometry (GC-MS) were used to detect 56 aroma compounds from mango, and the major components were monoterpenes (α-pinene, β-pinene, myrcene, etc.) and sesquiterpenes (α-copaene, α-gurjunene, trans-carophyllene, etc.) [26].

Chinese prickly ash, also called prickly ash, is one of the most economically important tree species in the Rutaceae family. Its cultivation history in China reaches back more than 2000 years, and it is widely distributed within Asia and around the world [27,28]. The dried peel has a unique fragrance and numbing flavor. It is usually added whole or as a powder in cooking and is widely used in Asia [29]. Especially in hot pot, prickly ash is one of the essential seasonings. In addition, fresh prickly ash is often used in cooking, but because it is inconvenient to store and transport and has a short shelf life, it is generally only available in small quantities during the harvest season [30]. Prickly ash is also used in Chinese medicine for its antidiarrheal, antiseptic, anti-inflammatory, and pain-relieving effects [31,32,33]. Aroma is one of the most important indicators of prickly ash fruit quality, and it is also the most direct basis by which consumers evaluate fruit quality. Differences in terpenoid content and composition are the main factors that cause fragrance differences among varieties. Prickly ash peel is rich in aromatic substances, including limonene, ocimene, phellandrene, thujone, linalool, terpineol, 4-terpene alcohol, and others [27,34]. Terpenoids give prickly ash its rich flavor and nutrition, as well as the potential for development into food, medicine, and cosmetics. Based on its color at commodity maturity, prickly ash is divided into green (*Zanthoxylum armatum*) and red prickly ash (*Zanthoxylum bungeanum*). Currently, *Z. armatum* is the main green prickly ash on the market; it is distributed primarily throughout southwestern China, including Sichuan, Guizhou, and Yunnan. Red prickly ash is mainly represented by *Z. bungeanum*. It is found in most regions of China, with the exception of the Northeast and Inner Mongolia, and its main production areas are concentrated in Shaanxi, Sichuan, Gansu, etc [35].

The two prickly ash types differ in their flavors and aromas, but the differences in their aroma components and key synthesis genes are unknown. Characterizing their terpenoid contents and the expression patterns of related genes will help to identify the underlying mechanisms responsible for the difference in aroma between green and red prickly ash. We therefore analyzed the transcriptome and metabolome of fruits from the two species at different developmental stages. Using methods such as weighted gene co-expression network analysis (WGCNA), redundancy analysis (RDA), and pathway enrichment analysis of differentially expressed genes, we characterized differences in aroma components and analyzed the expression patterns of terpenoid biosynthesis genes. Studying the aroma of prickly ash fruit will help to identify characteristic metabolites and key genes, thereby providing a molecular basis for the rapid improvement of prickly ash fruit quality.

## 2. Materials and Methods

### 2.1. Fruit Materials

*Z. bungeanum* (red prickly ash) and *Z. armatum* (green prickly ash) fruit samples from different developmental stages were collected at the Fengxian Prickly Ash Experimental Station of Northwest Agriculture and Forestry University (33°59′6.55′′ N 106°39′29.38′′ E). The fruits of red prickly ash were collected at the expansion stage (30 days after flowering, R1), the turning stage (60 days after flowering, R2), and the mature stage (90 days after flowering, R3). The fruits of green prickly ash were collected at the expansion stage (30 days after flowering, G1), the middle stage (60 days after flowering, G2), and the mature stage (90 days after flowering, G3). Three biological replicates were collected at each stage in 2020. The fruit samples were immediately frozen in liquid nitrogen and transferred to a −80 °C freezer for storage.

### 2.2. Analysis of Volatile Terpenoids in Prickly Ash Fruit by SPME-GC-MS

Solid-phase microextraction and gas chromatography-mass spectrometry (SPME-GC-MS) were used to detect terpenoid compounds in green and red prickly ash fruits from different developmental stages. The detection method was a slightly modified version of that described in Ma et al. [36]. The tissue was ground under liquid nitrogen, and 0.1 g of ground tissue was weighed out for analysis. A 65-µm polydimethylsiloxane (PDMS)/divinylbenzene (DVB) extraction head was used to collect volatile samples at 70 °C for 30 min, and a Thermo Scientific Trace 1310 GC-MS (Thermo Scientific, Waltham, MA, USA) was immediately used at 250 °C to analyze the samples. The column stationary phase was polysiloxane. Volatile components were separated on a TG-5MS column, and the heating program of the column oven was as follows: initial temperature 40 °C; temperature increased at 5 °C/min to 150 °C; temperature increased at 3 °C/min to 160 °C; temperature increased to 200 °C and maintained at 200 °C for 3 min; temperature increased at 20 °C/min to 300 °C and maintained at 300 °C for 10 min. Helium was used as the carrier gas, the flow rate was 80.0 mL/min, and the injection port temperature was maintained at 250 °C. The mass spectrometry conditions were ionization energy of 70 ev and a scanning mass range of 50–550 *m*/*z*. The NIST mass spectrum library was used to identify volatile organic compounds. Each sample was repeated three times. Linalool was used as an external standard and prepared in alcohol at seven concentrations (0.01, 0.02, 0.04, 0.08, 0.16, 0.32, and 0.64 mg/mL) to develop a standard curve for quantitative analysis of terpenoids. The resulting standard curve was y = 7 × 10^8^x + 2 × 10^7^, *R*^2^ = 0.953.

### 2.3. Analysis of Non-Volatile Terpenoids in Prickly Ash Fruit by UPLC-MS/MS

Fruits of green and red prickly ash from different developmental stages were vacuum freeze-dried in a lyophilizer (Scientz-100F, Ningbo Scientz Biotechnology Co., Ltd., Ningbo, China) and ground to powder with a grinder (MM 400, Retsch, Haan, Germany). 1.2 mL of 70% methanol extraction solution was added to 100 mg of sample, shaken, and placed in a refrigerator at 4 °C for 16 h. The sample was centrifuged at 12,000 rpm for 10 min, and the supernatant was filtered through a 0.22-μm microporous membrane and saved in a sample bottle for UPLC-MS/MS analysis (Ultra Performance Liquid Chromatography UPLC, Nexera X2, Shimadzu, Kyoto, Japan; Tandem mass spectrometry, Applied Biosystems 4500 QTRAP, Thermo Scientific). Metabolome analysis, including the identification and quantification of metabolites, was performed by Wuhan MetWare Biotechnology Co., Ltd. (Wuhan, China) by standard procedures [37,38]. The chromatographic column was an Agilent SB-C18 (1.8 μm, 2.1 mm × 100 mm), phase A was ultrapure water with 0.1% formic acid, and phase B was acetonitrile with formic acid). Analyst 1.6.1 software (AB SCIEX, Concord, ON, Canada) was used to analyze metabolite data, and principal component analysis (PCA) and orthogonal projections to latent structures discriminant analysis (OPLS-DA) were performed on data from all samples. Metabolites whose variable importance in projection (VIP) was ≥1 with *p* < 0.05 for the model variables were defined as differentially accumulated metabolites. the Kyoto Encyclopedia of Genes and Genomes (KEGG, www.genome.jp/kegg (accessed on 10 February 2021)) and the Plant Metabolic Network (PMN, www.plantcyc.org (accessed on 10 February 2021)) databases were used for pathway enrichment analysis of differentially abundant metabolites.

### 2.4. RNA Extraction and Sequencing

Total RNA was extracted from fruit samples using the TaKaRa MiniBEST Plant RNA Extraction Kit (TaKaRa, Beijing, China) according to the manufacturer’s instructions. A NanoDrop 2000 spectrophotometer (Thermo Scientific, Pittsburgh, PA, USA) was used to evaluate the purity, concentration, and integrity of the RNA samples. High-quality RNA was used to construct cDNA libraries, and the Illumina NovaSeq 6000 platform was used to sequence the resulting 150-bp paired-end reads. High-quality reads were obtained by trimming sequencing adaptors and removing low-quality reads from the raw sequencing data, clean reads were mapped to the assembled transcripts, and gene expression levels were calculated as Fragments Per Kilobase per Million (FPKM).

### 2.5. Transcriptome Analysis

The DESeq R package was used to identify differentially expressed genes (DEGs) in each sample group based on a *p*-value of <0.05 and a fold change of ≥2.0. The following databases were used for gene function annotation and pathway analysis: NR (NCBI non-redundant protein sequence), NT (NCBI non-redundant nucleotide sequence), Pfam (protein family), KOG/COG (Clusters of Orthologous Groups of proteins), Swiss-Prot (manually annotated and commented protein sequence), KO (KEGG Ortholog), and GO (Gene Ontology).

### 2.6. Weighted Gene co-Expression Network Analysis (WGCNA)

WGCNA analysis was performed on the transcriptome data to identify highly coordinated gene sets during fruit development in green and red prickly ash, as well as associations between gene sets and aroma. Genes with FPKM > 1 were used to construct a weighted gene co-expression network and partition module with the WGCNA v1.6.6 package in R v3.4.4. The blockwiseModules function was used to build a scale-free network, and all parameters were set to default values [39].

### 2.7. Redundancy Analysis (RDA)

To identify genes that played a key role in terpene synthesis during fruit development, the terpenoid contents at different stages and the expression levels of terpenoid synthesis-related genes were analyzed by redundancy analysis (RDA). RDA is the PCA analysis of the fitted value matrix of a multiple linear regression between the response variable matrix and the explanatory variable. Online analysis software in the GENE DENOVO cloud platform was used to perform RDA analysis on terpenoids and terpene synthesis-related genes of prickly ash (https://www.omicshare.com/ (accessed on 10 February 2021)).

### 2.8. Statistical Analyses

The experimental data were analyzed using SPSS 23.0 (IBM, Armonk, NY, USA), and significant differences were determined by ANOVA and Duncan’s multiple range test (*p* < 0.05).

## 3. Results

### 3.1. Terpenoid Metabolites of Red and Green Prickly Ash at Three Developmental Stages

To study the synthesis and accumulation of terpenoids in fruits, the total terpenoid contents of fruits from different developmental stages of green and red prickly ash (Figure 1a) were measured by gas chromatography-mass spectrometry (GC-MS). Sixty-one terpenoids were identified from red prickly ash and 41 from green prickly ash (Appendix A). Sixteen terpenoids were also detected in prickly ash fruit using LC-MS (Appendix A).

In the early stage of fruit development (R1), the total terpenoid content of red prickly ash was 193.34 mg/100 g (Figure 1b). In the middle stage (R2), it rose to 297.93 mg/100 g, and at the mature stage, it dropped to 238.51 mg/100 g. The total terpenoid content of green prickly ash fruit rose from the early stage of development (G1) to 419.39 mg/100 g at the mature stage (G3). The total terpenoid content was significantly higher (1.73-fold) in green prickly ash than in red prickly ash. Piperitone was the most abundant terpenoid in green prickly ash fruit, whereas limonene was the most abundant in red prickly ash. Twenty-two monoterpenoids, 34 sesquiterpenes, four diterpenes, and one tetraterpene were detected in red prickly ash; 12 monoterpenes, 23 sesquiterpenes, two diterpenes, two triterpenes, and two tetraterpenes were detected in green prickly ash (Figure 1c and Figure 2a). Twenty-seven terpenoids were unique to red prickly ash, including limonene, nerol, and neral. Seven terpenoids were unique to green prickly ash, including (-)-myrtenol and caryophyllene oxide. Orthogonal projections to latent structures discriminant analysis (OPLS-DA) was performed on all samples.

The abscissa represents the difference between groups, and the ordinate represents the difference within groups (Appendix A). OPLS-DA results showed that the differences between the sample groups were large, whereas the differences within the groups were small. The percentage of variation explained by PC1 and PC2 in the principal components analysis (PCA) was 90.20% (PC1 + PC2), and these two PCs therefore represented the main characteristics of the sample (Appendix A).

Green and red prickly ash could be divided into two groups based on the terpenoid index, indicating that the two prickly ash types differed significantly in terpenoid composition and content. Monoterpenes and sesquiterpenes were the main terpenoid components of green and red prickly ash fruit, accounting for more than 99.9% of the total terpenoid content (Figure 2b,c). By contrast, diterpenoids, triterpenoids, and tetraterpenoids accounted for less than 0.01% of the total content. The proportion of different terpenoids changed over the course of fruit development. During the development of green prickly ash fruit, the percentage of monoterpenoids rose from 44% to 58%, and that of sesquiterpenes dropped from 56% to 42%. The percentage of monoterpenoids in red prickly ash rose from 67% to 73%, and that of sesquiterpenes dropped from 33% to 27%.

In general, the number of different terpenoids was higher in red than in green prickly ash fruit, but the total terpenoid content was significantly higher in green than in red fruit. Monoterpenoids and sesquiterpenoids were the main aroma components of prickly ash; the proportion of monoterpenoids increased and that of sesquiterpenes declined over the course of fruit development. Different terpenoid components and differences in relative terpenoid abundance were the main reasons for the difference in aroma between green and red prickly ash.

### 3.2. Differentially Expressed Genes in Red and Green Prickly Ash Fruit at Three Developmental Stages

To study the molecular regulatory mechanisms of aroma substances during prickly ash fruit development, we performed transcriptome sequencing on green and red fruits at three developmental stages. Together, the 18 transcriptome samples produced 117.42 Gb of clean data, more than 5.71 Gb per sample, with a percentage of Q30 bases above 91.88%. The clean reads were compared and annotated, and 64,999 genes were obtained, 8036 of which were newly discovered. There were 2569 DEGs between different developmental stages of green prickly ash fruit, 135 of which were differentially expressed among all stages (Figure 3a). There were 11,349 DEGs between different developmental stages of red prickly ash fruit, 359 of which were differentially expressed among all stages (Figure 3b).

To analyze the functions of DEGs during fruit development, KEGG enrichment analysis was performed on DEGs between the early (G1 and R1) and ripe fruit stage (G3 and R3) of the two species (Figure 3c). The DEGs between early and ripe green fruits were enriched mainly in starch and sucrose metabolism (ko00500), phenylpropanoid biosynthesis (ko00940), flavonoid biosynthesis (ko00941), phenylalanine metabolism (ko00360), and ubiquinone and other terpenoid-quinone biosynthesis (ko00130). The DEGs between early and ripe red fruits were enriched mainly in starch and sucrose metabolism (ko00500), biosynthesis of amino acids (ko01230), carbon metabolism (ko01200), phenylalanine metabolism (ko00360), and terpenoid backbone biosynthesis (ko00900) (Figure 3d). We also performed KEGG enrichment analysis on DEGs between G3 and R3 to study differences in gene expression between the two varieties at the mature stage (Figure 3e). These DEGs were enriched mainly in glycolysis/gluconeogenesis, terpenoid backbone biosynthesis, and flavonoid biosynthesis pathways. It should be noted that glyceraldehyde-3P, synthesized by the glycolysis/gluconeogenesis pathway, is the initial substrate of the terpenoid backbone biosynthesis pathway [40]. Many physiological processes are involved in the development of prickly ash fruit, including the synthesis of starch, sucrose, and amino acids. A large number of DEGs were related to terpenoid synthesis, which is the metabolic pathway that underlies the aroma difference between green and red prickly ash fruits. Furthermore, the difference in color between green and red prickly ash fruits is due mainly to differences in gene expression in the flavonoid biosynthesis pathway.

### 3.3. Identification of a WGCNA Module Related to Aroma Substances during Fruit Development of Green and Red Prickly Ash

To investigate the molecular mechanism of aroma synthesis during fruit development in green and red prickly ash, we used weighted gene co-expression network analysis (WGCNA) to identify modules with high gene co-expression. WGCNA divided the genes into 25 modules (Figure 4a,b).

There were over 1000 genes in the darkseagreen3, antiquewhite4, firebrick2, paleturquoise, and brown modules, whereas there were fewer than 100 genes in the sienna1, whitesmoke, antiquewhite2, lightslateblue, and deeppink1 modules. The darkseagreen3 module contained genes related to amino acid synthesis, plant hormone signal transduction, and carbon metabolism. These genes were negatively correlated with green prickly ash and positively correlated with red prickly ash. The brown module contained genes related to genetic information processing, including ribosome, RNA transport, spliceosome, and protein processing in endoplasmic reticulum. Genes from these pathways were positively correlated with green prickly ash and negatively correlated with red prickly ash (Figure 4c). The antiquewhite4 and firebrick2 modules were negatively correlated with phenylpropanoid biosynthesis and starch and sucrose metabolism.

Genes related to terpenoid synthesis were concentrated mainly in the paleturquoise (14 genes), coral3 (13), and navajowhite1 (7) modules. These three modules had a high positive correlation with the early stage of fruit development in red prickly ash (R1) and a negative correlation with the early stage of fruit development in green prickly ash (G1) (Figure 4d). The results showed that there were differences in the synthesis of secondary metabolites and plant hormone-related signal transduction between green and red prickly ash. Moreover, differences in the expression patterns of genes related to terpene synthesis in green and red fruits may be the main reason for the differences in their aroma components and contents.

### 3.4. Expression Patterns of Genes Related to the Terpene Synthesis Pathway

The synthesis of terpenes occurs primarily through the mevalonate (MVA) and 2-C-methyl-D-erythritol-4-phosphate (MEP) pathways [41]. To further investigate the molecular mechanisms underlying aroma differences between green and red prickly ash, we analyzed the expression patterns of 41 genes related to terpene synthesis in the MVA and MEP pathways (Figure 5, Appendix A). Reactions of the MVA pathway occur mainly in the cytoplasm, beginning with the condensation of acetyl CoA. Two acetyl CoA are condensed in a reaction catalyzed by acetyl-CoA C-acetyltransferase (ACAT) to form acetoacetyl CoA, and acetoacetyl CoA and acetyl CoA are then condensed by 3-hydroxy-3-methyl-glutaryl-CoA synthase (HMGS) to produce 3-hydroxy-3-methylglutaryl CoA (HMG-CoA). HMG-CoA is reduced to mevalonic acid in a reaction catalyzed by 3-hydroxy-3-methyl-glutaryl-CoA reductase (HMGR).

Mevalonate is acted upon successively by mevalonate kinase (MVK), phosphomevalonate kinase (PMK), and mevalonate-5-pyrophosphate decarboxylase (MVD) to form isopentenyl diphosphate (IPP). IPP and dimethylallyl diphosphate (DMAPP) can be interconverted through the action of isopentenyl-diphosphate delta-isomerase (IDI). Two IPP and one DMAPP undergo a two-step condensation reaction to produce farnesyl diphosphate (FPP). FPP can be used to synthesize sesquiterpenoids and triterpenoids through the activities of various enzymes. For example, germacrene D synthase (GDS) catalyzes the production of germacrene (a sesquiterpene) from FPP, and lupeol synthase (LUP) catalyzes the formation of lupeol (a triterpenoid) from FPP.

The MEP pathway takes place in the plastids, including chloroplasts, chromoplasts, and leucoplasts. Pyruvate and glyceraldehyde-3-phosphate are the initial precursors of the MEP pathway. They are acted upon by 1-deoxy-D-xylulose 5-phosphate synthase (DXS) to form 1-deoxy-D-xylulose 5-phosphate (DXP). DXP is converted to 2-C-methyl-D-erythritol 4-phosphate (MEP) by 1-deoxy-D-xylulose-5-phosphate reductoisomerase (DXR). MEP is acted upon by 2-C-methyl-D-erythritol 4-phosphate cytidylyltransferase (MCT) to form 4-diphosphocytidyl-methylerythritol (CDP-ME), and CDP-ME is transformed into 1-hydroxy-2-methyl-2-butenyl-4-diphosphate (HMBPP) by 4-diphosphocytidyl-2-C-methyl-D-erythritol kinase (CMK) and 1-hydroxy-2-methyl-2-butenyl 4-diphosphate synthase (HDS). Finally, 1-hydroxy-2-methyl-2-butenyl 4-diphosphate reductase (HDR) catalyzes the formation of IPP and DMAPP from HMBPP. IPP and DMAPP are condensed to form geranyl diphosphate (GPP) by farnesyl diphosphate synthase/geranyl diphosphate synthase (FDPS/GPS), and GPP is the biosynthetic precursor of monoterpenoids, ubiquinone, and other terpene quinones. For example, linalool synthase (LIS) uses GPP to form linalool (a monoterpenoid), and monoterpenes are the main components of prickly ash terpenoids. GPP is used to produce geranylgeranyl diphosphate (GGPP) through multiple reactions, and GGPP is the precursor of diterpenoids, tetraterpenoids, ubiquinone, and other terpenoid quinones.

We measured the expression of 41 genes in the terpene biosynthesis pathway throughout the development of green and red prickly ash fruit and found that their expression patterns were quite different (Figure 5). Some genes had low expression in green prickly ash and high expression in red prickly ash (e.g., *HMGR2*, *IPI/IDI2*, and *IPI/IDI2*), whereas others showed the opposite pattern (e.g., *MVK1* and *MVK2*). During the development of green prickly ash, most terpene synthesis genes had the highest expression levels at the middle stage of fruit development (G2), suggesting that this is the key stage for terpenoid synthesis in green prickly ash. In mature red prickly ash fruit (R3), all terpene-related genes were downregulated except for LIS, suggesting that the rate of terpenoid synthesis is lower during the mature period in red fruit.

### 3.5. Correlations Between Terpenoid Contents and Expression of Terpenoid Synthesis Genes

We conducted intergroup correlation analysis and RDA analysis on terpenoid content and terpene synthesis-related gene expression during the development of prickly ash fruit to identify key genes in terpenoid synthesis (Figure 6). The intergroup correlation heat map showed differences in the correlations between specific genes and terpenoids, indicating that individual genes have different functions in terpenoid synthesis.

Piperitone, the terpenoid with the highest content in green prickly ash fruits, had the highest correlations with *HDS2*, *MVK2*, and *MVD*, followed by *HMGR1*, *DXS1*, and *GDS2*. α-Pinene, which had the second highest content, also had high correlations with the same genes. Limonene, the terpenoid with the highest content in red prickly ash fruit, had the highest correlations with *FDPS2* and *FDPS3*.

To further investigate key genes involved in terpenoid synthesis, we performed RDA on Piperitone and α-pinene, the main monoterpenoid components of green prickly ash, had a significant positive correlation with the expression of *HDS2*, *MVK2*, *DXS1*, *DXR2*, and *MVD*, indicating that these genes play an active role in the synthesis of green prickly ash monoterpenes (Figure 7). β-Elemene is the main sesquiterpene in green prickly ash, and it had a significant positive correlation with the expression of *MVD*, *HMGR1*, and *GDS1*, indicating that these genes are important for sesquiterpene synthesis. In red prickly ash, *FDPS2*, *FDPS3*, and *DXS2* had the highest correlations with the most abundant terpenoid, limonene, indicating that they may be the key genes for terpenoid synthesis. Differences in terpenoid composition and content are the main factor underlying aroma differences between green and red prickly ash, and differences in terpenoid synthesis-related gene expression are the molecular basis for these aroma differences.

In green prickly ash, genes whose expression was highly correlated with the main terpenoids were found in the MVA and MEP pathways (e.g., *HMGR1*, *MVK2*, *DXS1*, *HDS2*, etc.). By contrast, genes whose expression was highly correlated with the main terpenoids in red prickly ash came mainly from the MEP pathway (e.g., *HDS1*, *FDPS*, etc.). These results suggest that terpenoid synthesis occurs through both the MVA and MEP pathways in green prickly ash but primarily through the MEP pathway in red prickly ash.

## 4. Discussion

Aroma substances are an important class of volatile fragrance components that are widely distributed in higher plants [42]. The unique aromas of different plants are determined by specific types of aromatic substances. For example, the characteristic aromatic substances of citrus are terpenes, those of apple are esters, and that of strawberry is furanone [43,44,45]. Aromas not only impart a specific flavor to fruit but also have diverse biological functions and play a role in information transmission during interactions among plants, microorganisms, insects, and the environment. Terpenoids are one of the most abundant components in plant aromas and are important indicators of fruit ripeness and quality [46].

Here, we analyzed the transcriptome and metabolome of prickly ash fruits at different developmental stages in order to characterize the metabolites and molecular synthesis mechanisms underlying the aromas of green and red fruit. GC-MS results showed that the total terpenoid content of green prickly ash reached 419.39 mg/100 g at the maturity stage (G3), whereas that of red prickly ash reached a maximum of 297.93 mg/100 g at the middle developmental stage (R2). Piperitone was the main terpenoid component of green prickly ash fruit, whereas that of red prickly ash fruit was limonene. Citrus is also an economically important tree species in the Rutaceae family, and the main terpenoids in its peel are limonene and myrcene, the same as those of red prickly ash [47]. The content and composition of terpenoids showed that the aroma of red prickly ash is more similar to that of citrus than is the aroma of green prickly ash. In addition, the proportion of different terpenoids changes dynamically over the course of fruit development. Specifically, the proportion of monoterpenoids gradually increases, and that of sesquiterpenes gradually decreases. In summary, the aroma differences between green and red prickly ash are due to differences in the proportions of the main terpenoids and total terpenoid content.

Plant terpenoids are synthesized through the MVA and MEP pathways and finally by the catalysis of terpene synthase (TPS) [48]. To clarify the molecular mechanisms underlying aroma differences between green and red prickly ash, we analyzed the expression patterns of 41 terpene synthesis-related genes in the MVA and MEP pathways. Terpene synthesis genes had different expression patterns in different developmental stages of green and red prickly ash fruits. In green prickly ash, most genes in the MVA and MEP pathways had the highest expression in the middle stage of fruit development, indicating that G2 is the key stage for terpene synthesis in green prickly ash. In the mature stage of red prickly ash, all genes except *LIS* showed decreased expression, indicating that terpenoid synthesis declines at the mature stage in red prickly ash. Correlation analysis and RDA analysis of terpenoid composition, terpenoid content, and expression of terpenoid-related genes showed that *HDS2*, *MVK2*, and *MVD* are the key genes for terpene synthesis in green prickly ash, whereas *FDPS* and *DXS* are the key genes for the synthesis of terpenoids in red prickly ash. Differences in the expression patterns of terpene synthesis-related genes appears to be the underlying reason for the aroma differences between green and red prickly ash. *Atractylodes chinensis* is a Chinese medicinal plant from the Compositae family that is rich in terpenoids. Real-time quantitative PCR results showed that *FPPS*, *HMGR*, and *DXS* were expressed in different organs of *A. chinensis* and were key genes for terpenoid synthesis. Similarly, expression of *DXS* and *HDR* genes was positively correlated with monoterpenoid content during grape ripening, and they were the key genes for grape terpenoid synthesis [49]. *FPPS*, *HMGR, DXS*, and *HDR* also played a role in the terpenoid synthesis of prickly ash fruit, indicating that terpenoid synthesis pathways and key genes are conserved among different species [50].

In addition to providing aromas and serving signaling functions in plants, terpenoids and their derivatives also have significant potential for commercial development and are widely used in industry, food, cosmetics, and medicine [51]. For example, myrcene, limonene, and linalool are the most common terpenes in plants and are widely used as additives and flavoring agents in perfumes, condiments, and cosmetics. Terpenoids also have medicinal value and health effects. Paclitaxel is a highly effective anti-cancer drug that can inhibit the expansion of tumor cells, and artemisinin is a specific anti-malarial drug [52,53]. Lycopene, a tetraterpene compound, can effectively inhibit lipid oxidation and scavenge hydroxyl free radicals; it can also prevent cardiovascular and cerebrovascular diseases and delay aging [54,55]. Some terpenoids can also serve as environmentally friendly herbicides and pesticides in agriculture [56]. Multiomics has become an important method for studying secondary metabolism and can be used to address biological problems from multiple angles. Here, transcriptomics and metabolomics were used to study terpenoid synthesis at different developmental stages of prickly ash fruit, revealing differences in aroma components of green and red fruit and providing insight into their synthesis mechanisms. These results provide a basis for further development of terpenoids in prickly ash fruit.

## Figures and Tables

**Figure 1 foods-10-00391-f001:**
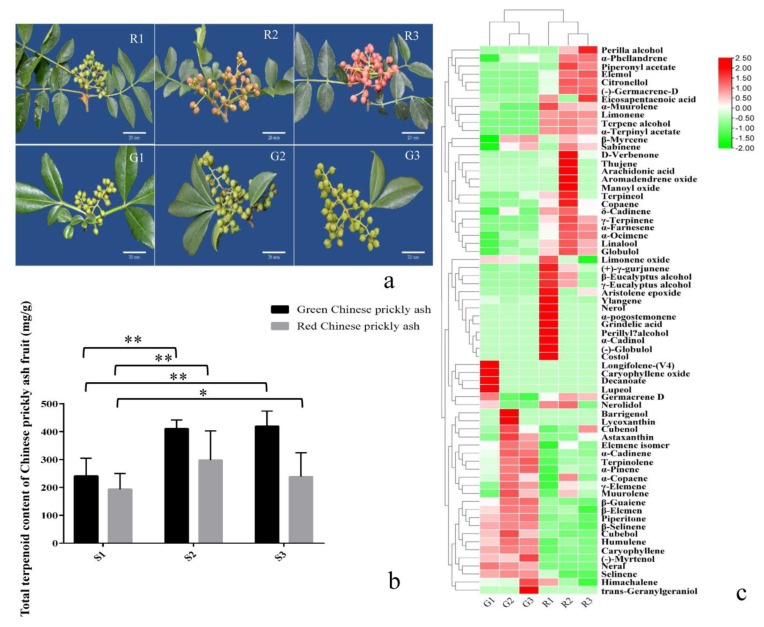
Terpenoid contents of green and red prickly ash fruits at different developmental stages. (**a**) Green and red prickly ash fruits at different developmental stages. The scale bar is 20 mm. (**b**) The total terpenoid content in green and red prickly ash fruits at different developmental stages. * 0.01 < *p* < 0.05, ** *p* < 0.01. (**c**) Heat map of individual terpenoid contents in green and red prickly ash fruits at different developmental stages.

**Figure 2 foods-10-00391-f002:**
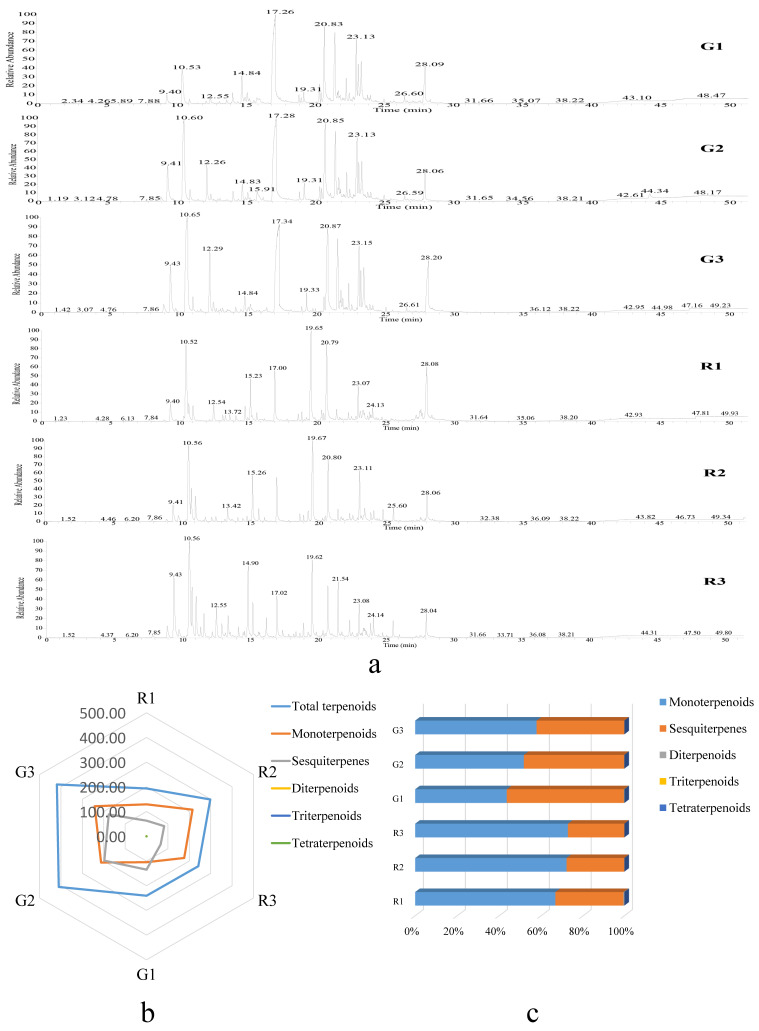
GC-MS analysis of terpenoids in green and red prickly ash fruits. (**a**) GC-MS chromatograms of different developmental stages of green and red prickly ash fruit. (**b**) Radar chart of the contents of different terpenoid classes in green and red prickly ash fruits at three developmental stages (mg/100 g). (**c**) Percentage of different terpenoid classes in green and red prickly ash fruit at different developmental stages.

**Figure 3 foods-10-00391-f003:**
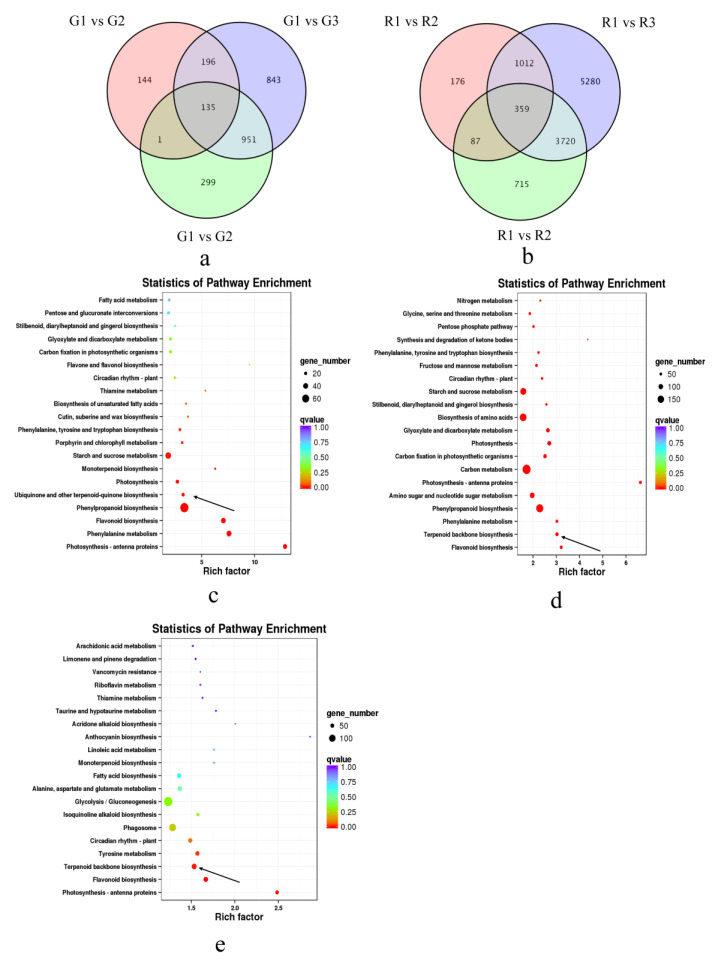
Analysis of differentially expressed genes at different developmental stages of green and red prickly ash. (**a**) Venn diagram of differentially expressed genes at three developmental stages of green prickly ash. (**b**) Venn diagram of differentially expressed genes at three developmental stages of red prickly ash. (**c**) KEGG enrichment analysis of differentially expressed genes at two developmental stages of green prickly ash. (**d**) KEGG enrichment analysis of differentially expressed genes at two developmental stages of red prickly ash. (**e**) KEGG enrichment analysis of differentially expressed genes at the mature stage of green and red prickly ash.

**Figure 4 foods-10-00391-f004:**
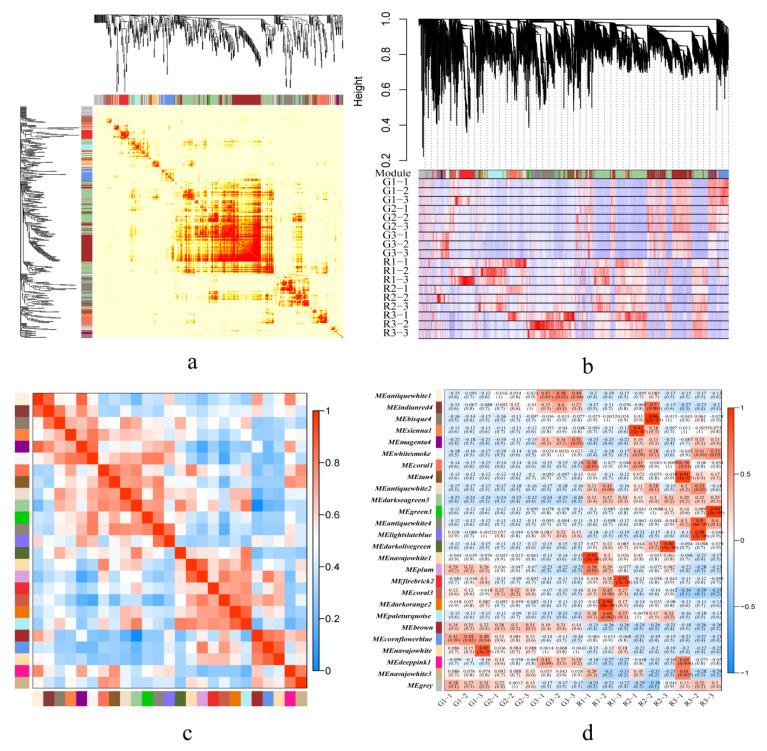
Weighted gene co-expression network analysis. (**a**) Network heatmap plot. (**b**) Cluster dendrogram. (**c**) Eigengene adjacency heat map. (**d**) Module-trait relationships (upper values are the correlation coefficients, and lower values are the *p*-values).

**Figure 5 foods-10-00391-f005:**
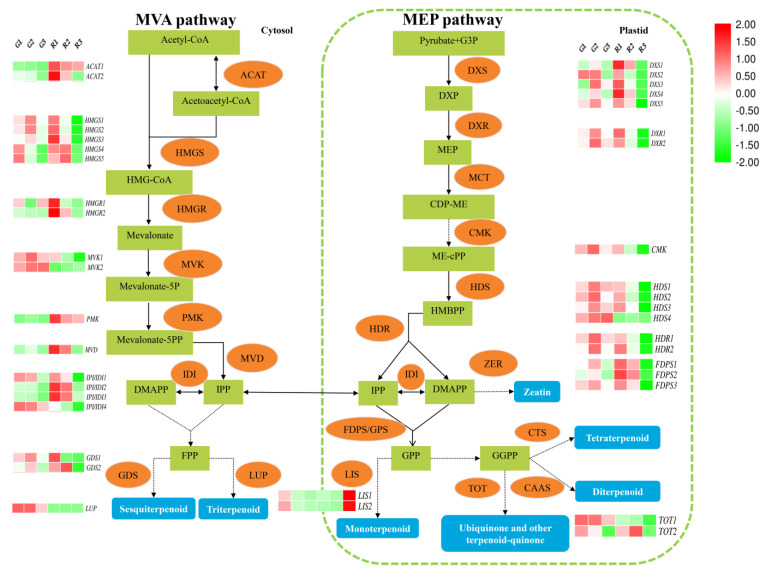
Expression patterns of terpene synthesis-related genes at different developmental stages of green and red prickly ash fruit. The red boxes represent genes and the green boxes represent metabolites.

**Figure 6 foods-10-00391-f006:**
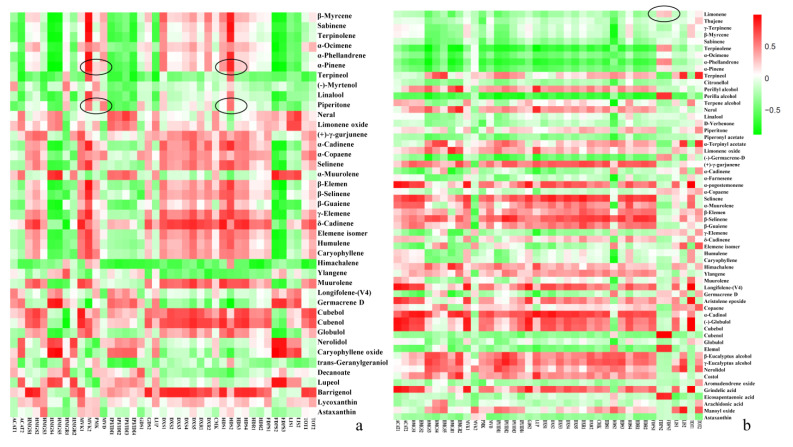
Intergroup correlation analysis of terpenoids and terpenoid synthesis-related genes in green and red prickly ash. (**a**) Intergroup correlation analysis of terpenoids and terpenoid synthesis-related genes in green prickly ash fruit. (**b**) Intergroup correlation analysis of terpenoids and terpenoid synthesis-related genes in red prickly ash fruit.

**Figure 7 foods-10-00391-f007:**
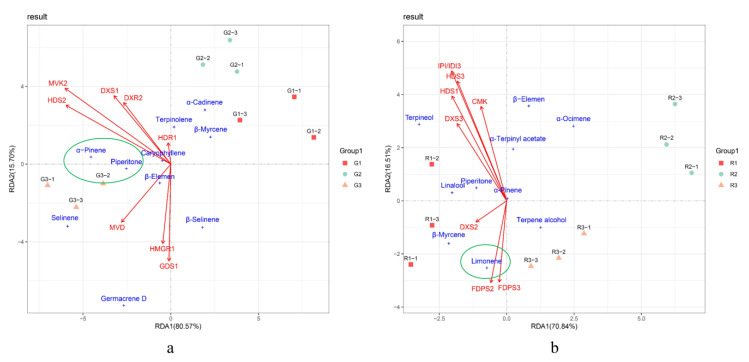
Redundancy analysis (RDA) of terpenoids and terpenoid synthesis genes in prickly ash fruit. (**a**) RDA analysis of terpenoids and terpenoid synthesis genes at different developmental stages of green prickly ash. (**b**) RDA analysis of terpenoids and terpenoid synthesis genes at different developmental stages of red prickly ash.

## Data Availability

The datasets generated for this study are available on request to the corresponding author.

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
