# Peer review of "Transcriptome and Metabolome Dynamics Explain Aroma Differences between Green and Red Prickly Ash Fruit"

_foods, 2021, doi:10.3390/foods10020391_

Round 1
Reviewer 1 Report
Manuscript foods-1081814 presents terpenoid metabolomic profiles using GC-MS and HPLC-MS of the fruits of Zanthoxylum bungeanum (red prickly ash) and Z. armatum (green prickly ash) that contribute to the different aromas of these two species. In addition, the authors have correlated the major terpenoids with gene expression in the developing fruits.
Suggestions for the authors:
Abstract: Please include the scientific names of the plants.
Keywords: Avoid repeating keywords that are already in the title (i.e., omit prickly ash, transcriptome, metabolome). Include the scientific names of the plants and other keywords that researchers are likely to use in a search.
Introduction:
The introduction to terpenoid synthesis is a welcome addition, especially for readers of Foods who may not be familiar with terpenoids.
Lines 60-61: “In sweet orange, isopentenyl diphosphate and dimethylallyl diphosphate are the main aroma components of fruit…” This is not true. I would suggest ethyl butanoate and maybe myrcene for the fruit and limonene and myrcene for the peel. See: Moufida et al., Phytochemistry 2003, 62, 1283-1289; Qiao et al., Molecules 2008, 13, 1333-1344; Dosoky et al., Int. J. Mol. Sci. 2018, 19, 1966; Feng et al., J. Agric. Food Chem. 2018, 66, 203−211.
The introduction is incomplete in that the reader does not get any information about the plants. The introduction should include background information on these two species. Include the complete scientific name (including the botanical authority), the native ranges for each, and the varieties or cultivars for each.
Materials and Methods:
Line 111: Please define PDMS/DVB
Section 2.3.: Should this be “Analysis of non-volatile terpenoids in prickly ash fruit by UPLC-MS/MS” ?
Results: Lines 180-181 and lines 195-200: The GC-MS analyses of all of the samples of both species should be presented in a table. This will be beneficial to readers to for ready comparison between developmental samples and species.
Figures 1c, 2a, 3, 4, and 6 are impossible to read and should be replaced.
Author Response
We are very grateful to the reviewers and chief editor for their constructive comments and suggested amendments. Their inputs have helped to improve the paper tremendously. We have carefully studied the reviews and have revised our manuscript accordingly. We include our detailed responses to the reviewers. Please note that the comments from the reviewers are in italics followed by our responses in blue-inked text.
Reviewer 1
Abstract:
Comments 1: Please include the scientific names of the plants.
[Response] Corrected.
Keywords:
Comments 2: Avoid repeating keywords that are already in the title (i.e., omit prickly ash, transcriptome, metabolome). Include the scientific names of the plants and other keywords that researchers are likely to use in a search.
[Response] Corrected.
Introduction:
Comments 3: The introduction to terpenoid synthesis is a welcome addition, especially for readers of Foods who may not be familiar with terpenoids. Lines 60-61: “In sweet orange, isopentenyl diphosphate and dimethylallyl diphosphate are the main aroma components of fruit…” This is not true. I would suggest ethyl butanoate and maybe myrcene for the fruit and limonene and myrcene for the peel. See: Moufida et al., Phytochemistry 2003, 62, 1283-1289; Qiao et al., Molecules 2008, 13, 1333-1344; Dosoky et al., Int. J. Mol. Sci. 2018, 19, 1966; Feng et al., J. Agric. Food Chem. 2018, 66, 203−211. The introduction is incomplete in that the reader does not get any information about the plants. The introduction should include background information on these two species. Include the complete scientific name (including the botanical authority), the native ranges for each, and the varieties or cultivars for each.
[Response] Thank you for affirming the manuscript and recommending suitable documents for us. We corrected the inaccurate sentences in the introduction. At the same time, we added the complete scientific names of the two plants and added relevant background information.
Materials and Methods:
Comments 4: Line 111: Please define PDMS/DVB
[Response] Corrected.
Comments 5: Section 2.3.: Should this be “Analysis of non-volatile terpenoids in prickly ash fruit by UPLC-MS/MS”?
[Response] Corrected.
Results:
Comments 6: Lines 180-181 and lines 195-200: The GC-MS analyses of all of the samples of both species should be presented in a table. This will be beneficial to readers to for ready comparison between developmental samples and species.
[Response] Thank you for your valuable comments. In order to facilitate readers to compare the differences between samples, we added the GC-MS data of the two plants in Table S1 in the form of a table.
Comments 7: Figures 1c, 2a, 3, 4, and 6 are impossible to read and should be replaced.
[Response] We have modified the figures in the manuscript and added high-resolution pictures to the manuscript.
Reviewer 2 Report
Manuscript Number: foods-1081814
Title: Transcriptome and metabolome dynamics explain aroma differences between green and red prickly ash fruit
The paper deals with the evaluation of different of two species of prickly ash fruits at different developmental stages using transcriptome and metabolome approach.
The paper is well written and analyze the transcriptome and metabolome of prickly ash fruits at different developmental stages in order to characterize the metabolites and molecular synthesis mechanisms underlying the aromas of green and red fruit. The experimental design is well performed and described, and the subject is relevant to the aim and scope of the journal. Moreover, English language editing must be performed. Authors are invited to revise the manuscript according to the comments and recommendation listed below:
Introduction: lines 78-80: Insert reference/references to support.
Materials and methods: lines 129-135: Please describe the HPLC or UPLC equipment used for the analysis of volatile terpenoids; No reference to the type of the stationary phase of the column. Please add. The authors in this section did not show the univariate statistical analysis approach used and the significance. Instead in the results (Figure 1 b) showed the error standard and the significance. Please insert
Line 183-185 – The compounds identified using LC-MS must be commented later GC-MS analysis later to avoid confusion.
Results: Line 183 - please insert the reference to Figure 1 in the text.
Line 185 – The terpenoid metabolites identified using LC-MS is very important. Authors should add a table (Table S1) reporting all the individual data and relative statistical analysis. The authors used Principal Component Analysis (PCA) and Orthogonal Projections Discriminant Analysis (OPLS-DA) for differentially accumulated metabolites. These multivariate statistical analyses not shown in the results. Please comment the PCA and OPLS-DA results.
Lines 192-211 - please showed the results referring to figure 2.
Fig. 1 please change in the caption b “terpene” with “terpenoid”. The caption c “heat map of individual terpenoid content” is unclear to visualize. The image quality is scarce.
Author Response
We are very grateful to the reviewers and chief editor for their constructive comments and suggested amendments. Their inputs have helped to improve the paper tremendously. We have carefully studied the reviews and have revised our manuscript accordingly. We include our detailed responses to the reviewers. Please note that the comments from the reviewers are in italics followed by our responses in bold text.
Reviewer 2
Manuscript Number: foods-1081814
Title: Transcriptome and metabolome dynamics explain aroma differences between green and red prickly ash fruit
The paper deals with the evaluation of different of two species of prickly ash fruits at different developmental stages using transcriptome and metabolome approach.
The paper is well written and analyze the transcriptome and metabolome of prickly ash fruits at different developmental stages in order to characterize the metabolites and molecular synthesis mechanisms underlying the aromas of green and red fruit. The experimental design is well performed and described, and the subject is relevant to the aim and scope of the journal. Moreover, English language editing must be performed. Authors are invited to revise the manuscript according to the comments and recommendation listed below:
[Response] Thank you for your professional comments and affirmation of the manuscript. Your comments will greatly help us improve the quality of the manuscript. We invited native English-speaking editors to edit the language of our manuscript. We revised and responded one-to-one based on your comments, and listed the responses below.
Introduction:
Comments 1: lines 78-80: Insert reference/references to support.
[Response] Corrected.
Materials and methods:
Comments 2: lines 129-135: Please describe the HPLC or UPLC equipment used for the analysis of volatile terpenoids; No reference to the type of the stationary phase of the column. Please add. The authors in this section did not show the univariate statistical analysis approach used and the significance. Instead in the results (Figure 1 b) showed the error standard and the significance. Please insert Line 183-185 – The compounds identified using LC-MS must be commented later GC-MS analysis later to avoid confusion.
[Response] We have added the UPLC-MS/MS equipment model and manufacturer information to the materials and methods. We also added the types of stationary phases and statistical analysis methods in Materials and Methods. To avoid confusion, we analyzed the LC-MS and GC-MS separately, and added the data obtained to Table S1 and Table S2. The manuscript mainly analyzes the volatile aroma components obtained by GC-MS.
Results:
Comments 3: Line 183 - please insert the reference to Figure 1 in the text.
[Response] Corrected.
Comments 4: Line 185 – The terpenoid metabolites identified using LC-MS is very important. Authors should add a table (Table S1) reporting all the individual data and relative statistical analysis. The authors used Principal Component Analysis (PCA) and Orthogonal Projections Discriminant Analysis (OPLS-DA) for differentially accumulated metabolites. These multivariate statistical analyses not shown in the results. Please comment the PCA and OPLS-DA results.
[Response] We sorted the terpenoid data detected by LC-MS and GC-MS into Table S1, Table S2, Figure S1, and Figure S2, and added Principal Component Analysis (PCA) and Orthogonal Projection Discriminant Analysis to the manuscript (OPLS-DA) description.
Comments 5: Lines 192-211 - please showed the results referring to figure 2.
[Response] To show the content of Figure 2 more clearly, we transferred the PCA analysis to Figure S1, supplemented the relevant content in the manuscript, and quoted the figure to the corresponding position in the text.
Comments 6: Fig. 1 please change in the caption b “terpene” with “terpenoid”. The caption c “heat map of individual terpenoid content” is unclear to visualize. The image quality is scarce.
[Response] We have corrected the wrong words and improved the quality of the image.
Reviewer 3 Report
The manuscript delas with a transcriptomic and metabolomic variation of genes expression that are linked to aroma of two species (Zanthoxylum bungeanum and Zanthoxylum armatum) at different developmental stages.
The study is original and meets the expectation of Foods.
Nevertheless, the manuscript focused on the molecular methods of analysis and lost the essential information and interest of the study. What a shame !
The manuscript is really limited
- The objectives are not presented
- The Gene expression studies generally use very sensitive methods. However, the variation in gene expression is dependent on the species of the variety according to the environmental conditions (biotic or abiotic). If analyses are carried out on one-year samples, as is the case in this study, their scope and repeatability are limited and the interest of the study has a limited lifespan.
- Statistical analyses should be presneted and some of the analyses done should be modified (figure 1b for example)
- There are no conclusive sentences on the interests of this study and potentials uses
Further remarks :
Why the scientific name of Prickly ash appers only in abreviated form and only in M&M section.
Please cite the whole scientific name in Abstract and in Introduction.
Keywords : please replace Prickly ash (cited in title) by Zanthoxylum bungeanum
Introduciton L34-47
Please i twill more appreciated if this part is really illustrated by a scheme of the biosynthesis pathway (a figure).
P3 L93 what authors mean by WGCNA analysis, RDA analysis ? please give the whole name of the analyses at the first citation.
Section 2.1. Fruit materials
If the authors used a phneologic scale, please identify. What R1, R2 G1, G2 mean ?
The year of samples collecte was not given !!!?
This is important. Moreover for genetic study (more important in transcriptomic studies) the climatic conditions are importante. Moreover, the fragarances compounds (volatils) are produced by plant in reaction to envronmental parameters. Therefore, if the samples were collected in one environement in one year, The results will have « short shelf life » .
Results
Figure 1 is not referenced in the text ? This also the case for all figures of this manuscript.
Usually results presented are in connexion with figures and tables cited in the text.
Figure 1b what * means in this figure ?
If the * means difference between teh two species at each developmental stage, this difference is wrong. Please verify all results.
Moreover statistical analyses are not presented in M&M
Figure 1c is very hard to read. Please provide a better quality figure.
Author Response
We are very grateful to the reviewers and chief editor for their constructive comments and suggested amendments. Their inputs have helped to improve the paper tremendously. We have carefully studied the reviews and have revised our manuscript accordingly. We include our detailed responses to the reviewers. Please note that the comments from the reviewers are in italics followed by our responses in bold text.
Reviewer 3
The manuscript delas with a transcriptomic and metabolomic variation of genes expression that are linked to aroma of two species (Zanthoxylum bungeanum and Zanthoxylum armatum) at different developmental stages. The study is original and meets the expectation of Foods. Nevertheless, the manuscript focused on the molecular methods of analysis and lost the essential information and interest of the study. What a shame! The manuscript is really limited. The objectives are not presented. The Gene expression studies generally use very sensitive methods. However, the variation in gene expression is dependent on the species of the variety according to the environmental conditions (biotic or abiotic). If analyses are carried out on one-year samples, as is the case in this study, their scope and repeatability are limited and the interest of the study has a limited lifespan. Statistical analyses should be presneted and some of the analyses done should be modified (figure 1b for example).There are no conclusive sentences on the interests of this study and potentials uses.
[Response] Thank you very much for your positive assessment. We have carefully read your constructive comments and suggestions and made detailed revision according to these comments in our revised manuscript point to point. Some mistakes also have been carefully corrected. We have supplemented the manuscript and revised the sentence, to improve the quality of the article under your guidance. The language has also been comprehensively edited by a native English speaking professional (Editor Sandy) to ensure the quality of our writing. Please find our point-by-point responses below. We hope that you will find our revisions satisfactory.
Further remarks:
Comments 1: Why the scientific name of Prickly ash appers only in abreviated form and only in M&M section. Please cite the whole scientific name in Abstract and in Introduction.
[Response] Corrected.
Comments 2: Keywords: please replace Prickly ash (cited in title) by Zanthoxylum bungeanum
[Response] Corrected.
Comments 3: Introduciton L34-47, Please i twill more appreciated if this part is really illustrated by a scheme of the biosynthesis pathway (a figure).
[Response] Thank you for your comments. In order to enable readers to have a full understanding of the biosynthesis pathway, before we describe the results, we highlight the two pathways of terpenoid synthesis in the form of images in the results section 3.4.
Comments 4: P3 L93 what authors mean by WGCNA analysis, RDA analysis? please give the whole name of the analyses at the first citation.
[Response] Corrected.
Comments 5: Section 2.1. Fruit materials
If the authors used a phneologic scale, please identify. What R1, R2 G1, G2 mean? The year of samples collecte was not given!!!? This is important.
[Response] We introduced the meaning of R1, R2, R3, G1, G2, and G3 in the ‘Fruit materials’. The fruits of red prickly ash were collected at the expansion stage (30 days after flowering, R1), turning stage (60 days after flowering, R2), and mature stage (90 days after flowering, R3). The fruits of green prickly ash were collected at the expansion stage (30 days after flowering, G1), middle stage (60 days after flowering, G2), and mature stage (90 days after flowering, G3). Also, we added the year of sample collection, which is all samples collected in 2020.
Comments 6: Moreover for genetic study (more important in transcriptomic studies) the climatic conditions are importante. Moreover, the fragarances compounds (volatils) are produced by plant in reaction to envronmental parameters. Therefore, if the samples were collected in one environement in one year, The results will have «short shelf life».
[Response] Thanks for your valuable comments. We very much agree with you that climatic conditions have a great influence on the synthesis of aromatic compounds. Therefore, in order to minimize the impact of environmental parameters on the test results, our samples are collected in a unified management test station under the same climatic conditions.
Results
Comments 7: Figure 1 is not referenced in the text? This also the case for all figures of this manuscript.
[Response] Thank you for your careful comments. We referenced all the pictures in the manuscript.
Comments 8: Usually results presented are in connexion with figures and tables cited in the text. Figure 1b what * means in this figure? If the * means difference between teh two species at each developmental stage, this difference is wrong. Please verify all results.
[Response] We did not clearly show the meaning of * in Figure 1b, so we reworked Figure 1b, and all the significance analysis is based on the S1 stage (the first stage of fruit development).
Comments 9: Moreover statistical analyses are not presented in M&M.
[Response] We have added statistical analyses in the materials and methods section.
Comments 10: Figure 1c is very hard to read. Please provide a better quality figure.
[Response] We have improved the quality of all the images in the manuscript.
Round 2
Reviewer 2 Report
In my opinion, following the revisions, the manuscript can be accepted for publication.
Reviewer 3 Report
This manuscript was greatly improved accoding to recommendations made by the reviewers.